# Effects of Oxygen Pressure on the Microstructures and Nanomechanical Properties of Samarium-Doped BiFeO_3_ Thin Films

**DOI:** 10.3390/mi14101879

**Published:** 2023-09-29

**Authors:** Chih-Sheng Gao, Sheng-Rui Jian, Phuoc Huu Le, Wu-Ching Chou, Jenh-Yih Juang, Huang-Wei Chang, Chih-Ming Lin

**Affiliations:** 1Department of Materials Science and Engineering, I-Shou University, Kaohsiung City 84001, Taiwan; endygao@gmail.com (C.-S.G.); srjian@gmail.com (S.-R.J.); 2Department of Fragrance and Cosmetic Science, College of Pharmacy, Kaohsiung Medical University, Kaohsiung 80708, Taiwan; 3Center for Plasma and Thin Film Technologies, Ming Chi University of Technology, New Taipei City 24301, Taiwan; phle@mail.mcut.edu.tw; 4Department of Electrophysics, National Yang Ming Chiao Tung University, Hsinchu 30010, Taiwan; wcchou957@nycu.edu.tw (W.-C.C.); jyjuang@nycu.edu.tw (J.-Y.J.); 5Department of Physics, National Chung Cheng University, Chia-Yi 62102, Taiwan; 6Department of Physics, National Tsing Hua University, Hsinchu 30013, Taiwan

**Keywords:** Sm-doped BiFeO_3_ thin films, PLD, XRD, SEM, nanoindentation

## Abstract

In this study, samarium (Sm-10at%)-doped BiFeO_3_ (SmBFO) thin films were grown on platinum-coated glass substrates using pulsed laser deposition (PLD) to unveil the correlation between the microstructures and nanomechanical properties of the films. The PLD-derived SmBFO thin films were prepared under various oxygen partial pressures (*P*_O2_) of 10, 30, and 50 mTorr at a substrate temperature of 600 °C. The scanning electron microscopy analyses revealed a surface morphology consisting of densely packed grains, although the size distribution varied with the *P*_O2_. X-ray diffraction results indicate that all SmBFO thin films are textured and preferentially oriented along the (110) crystallographic orientation. The crystallite sizes of the obtained SmBFO thin films calculated from the Scherrer and (Williamson–Hall) equations increased from 20 (33) nm to 25 (52) nm with increasing *P*_O2_. In addition, the nanomechanical properties (the hardness and Young’s modulus) of the SmBFO thin films were measured by using nanoindentation. The relationship between the hardness and crystalline size of SmBFO thin films appears to closely follow the Hall–Petch equation. In addition, the *P*_O2_ dependence of the film microstructure, the crystallite size, the hardness, and Young’s modulus of SmBFO thin films are discussed.

## 1. Introduction

Recently, the development of high-performance electronic devices utilizing additional properties of electrons, such as spins and orbitals, has attracted extensive research interest owing to the compelling challenges encountered by electron charge-based technologies. The multiferroic materials exhibit ferromagnetic (FM) and ferroelectric (FE) properties simultaneously, which potentially can offer extra handles in manipulating the electronic states and hence, pave the way for developing next-generation devices. For instance, it has been proposed that electronic devices made of thin films possessing both FM and FE properties simultaneously can potentially utilize the electric field to control the magnetic order or vice versa using the magnetic field to control the electric polarization state of the device. Such operation concepts not only may lead to a significant reduction in power consumption but also provide extra degrees of freedom in designing information storage and spintronics devices [1]. However, in nature, only a few materials have these kinds of exotic characteristics because the structural, chemical, and physical conditions leading to FM and FE properties are inherently very different and often even mutually exclusive. Among the numerous multiferroic systems, bismuth ferrite (BiFeO_3_; BFO) has attracted enormous interest because of its multiferroic nature, i.e., the co-existence of FE behavior with a high Curie temperature *T*_C_ ~1103 K and antiferromagnetic (AFM) behavior with a Néel temperature *T*_N_ ~643 K, as well as piezoelectric (ferroelastic) properties at room temperature [2,3].

Different processing methods have been adopted for preparing BFO-based thin films, namely the sol-gel process [4,5], chemical vapor deposition [6], magnetron sputtering [7,8], and pulsed laser deposition (PLD) [9]. PLD is a versatile technique that enables the fabrication of thin films with multi-element stoichiometry as well as diverse structures and morphologies. However, devices made of BFO thin films fabricated by using various deposition techniques largely exhibit a high leakage current and large coercive field, which have been the main limiting factors preventing them from widespread applications. To overcome these shortcomings, in addition to improving the film quality by modifying the film deposition methods and conditions, doping with various rare-earth elements (e.g., La^3+^ [10], Nd^3+^ [11], Dy^3+^ [12], Y^3+^ [13], and Sm^3+^ [14]) into the Bi site of the BFO thin films have been proposed and tried in order to improve the magnetic and ferroelectric characteristics. From the engineering applications point of view, rare-earth element doping would introduce chemical pressure and lattice distortion, which, in turn, can manipulate the volatility of Bi atoms and hence affect the ferroelectric and magnetic properties of BFO thin films [15]. Taking La^3+^ doping as an example, its ionic radius (1.16 Å) is quite similar to Bi^3+^ by (1.15 Å); thus, it should not introduce a substantial amount of distortion in the lattice. Nevertheless, it has been observed to result in higher remanent polarization and coercive field of rare-earth-doped BFO as compared to its counterparts doped with a rare-earth element of a smaller radius, such as Pr, Nd, Sm, and Ho [16]. In contrast, the substitution of Bi^3+^ with the smaller Sm^3+^ (its ionic radius: 1.08 Å) was reported to result in significant structural distortion and substantial improvement in magnetic and ferroelectric properties [15]. It is noted, however, that in the latter case [15], the investigated samples were synthesized through solid-state reaction, while in the former case [16], the samples studied were prepared using pulsed laser deposition (PLD), indicating that the ionic size alone might not be adequate to account for the reported results, and more comprehensive understandings are required.

In addition to studying the processing parameter-dependent magnetic and FE properties, other aspects, such as the mechanical properties, of the rare-earth-doped BFO thin films are also of great interest. In particular, all of the thin films have to encounter various extents of contact loadings during the fabricating and/or packaging processes, which may cause significant degradation in device performance. Thus, a precise assessment of the film’s nanomechanical properties is required in order to implement the rare-earth-doped BFO thin film as a structural/functional element used in a nanoscale device. It has been generally conceived that the mechanical responses obtained from thin films may substantially deviate from that of the bulk material. In view of the fact that most practical applications of functional devices are fabricated with thin films, it is urgently needed to carry out precise measurements of the mechanical properties of the rare-earth-doped BFO thin films. In this respect, nanoindentation is widely used to characterize nanomechanical properties of a variety of film/substrate systems due to its high efficiency and practical convenience. A typical load–displacement curve (*P*-*h* curve) obtained from nanoindentation measurement can be readily used to obtain the nanomechanical properties of the indented materials such as the hardness, elastic modulus [17,18,19,20], and the elastic/plastic deformation behaviors [21,22,23,24,25].

Among the rare-earth-doped BFO systems, the Sm-doped BFO (SmBFO, Bi_1−x_Sm_x_FeO_3_) thin film has received particular research attention due to its unique structural transition over a relatively narrow concentration range at room temperature. SmBFO has been reported to exhibit a phase transition from the ferroelectric rhombohedral phase to the antiferroelectric orthorhombic phase with an increasing Sm doping amount. The morphotropic phase boundary appears in the vicinity of x = 0.13~0.15. In particular, anomalous high values of the out-of-plane piezoelectric coefficient (~110 pm V^−1^) and an enhanced dielectric constant at x = 0.14 are reported in such a system [26]. On the other hand, by studying a series of Bi_1-x_Sm_x_FeO_3_ thin films (x = 0.05–0.16) deposited using PLD, it was found that the 10 at% Sm-doped BFO thin films could result in films with an extremely fine grain structure and smooth film/substrate interface, which, in turn, gave rise to the largest remanent ferroelectric polarization as compared to films prepared under similar conditions but with different Sm doping concentrations [27]. Following the interesting observations from a previous study [27], the present work focuses on investigating the nanomechanical properties of a series of 10 at %-doping SmBFO thin films grown on Pt-coated glass substrates by using PLD at the substrate temperature of 600 °C under various oxygen partial pressures (*P*_O2_). The effects of *P*_O2_ on the film’s microstructure, surface morphology, and mechanical properties are studied by means of X-ray diffraction (XRD), scanning electron microscopy (SEM), and nanoindentation techniques. This study provides insights into the correlation between the nanomechanical properties and the microstructures and crystallite sizes of the SmBFO thin films grown at various *P*_O2_.

## 2. Materials and Methods

Prior to depositing the SmBFO thin film, a 20 nm thick Pt layer was grown on a Corning glass substrate (code 1737) at room temperature using RF-magnetron sputtering, which was intended to serve as an underlayer for the SmBFO thin film growth as well as the bottom electrode for subsequent ferroelectric property characterizations (not reported here). It is interesting to note here that the coated Pt layer often exhibits a strong (111)-textured characteristic (see below), which may have prominent consequences on the SmBFO thin films grown on it afterward. The Bi_0.9_Sm_0.1_FeO_3_ target was prepared by using the solid-state reaction method. Briefly, Bi_2_O_3_, Fe_2_O_3_, and Sm_2_O_3_ powders with a nominal composition of Bi_0.9_Sm_0.1_FeO_3_ were mixed and ground by using the ball milling method using 30 zirconia balls. The ground powder mixture was loaded into steel dies, which were then uniaxially pressed into a 10 mm diameter disc. The disc was sintered at 800 °C for 24 h in air and then naturally cooled to room temperature. The sintered disc was used as the target for subsequent thin-film growth via PLD. A Nd:YAG pulsed laser with a laser wavelength of 355 nm, energy of 2.5 mJ, and a repetition rate of 5 Hz was employed to prepare the SmBFO thin films. For the PLD process, the substrate temperature was kept at 600 °C, while the O_2_ partial pressure varied between 10 and 50 mTorr. The thickness of the SmBFO thin films was approximately 300 nm.

The crystal structure of the SmBFO thin films was determined using X-ray diffraction (XRD, Panalytical X’Pert diffractometer, Malvern Panalytical, Malvern, UK) with the Cu Kα radiation (λ = 1.5406 Å). Surface morphology and film thickness were examined using scanning electron microscopy (SEM, Hitachi S-4700, Tokyo, Japan) through plane-view and cross-sectional images. The nanoindentation measurements were conducted at room temperature with a Berkovich diamond indenter (whose radius of curvature is 50 nm) using the MTS NanoXP^®^ system (MTS Corporation, Nano Instruments Innovation Center, Oak Ridge, TN, USA). The resolutions of the loading force and displacement are 50 nN and 0.1 nm, respectively. All of the indentation depths were below 70 nm, and the strain rate was varied from 0.01 to 1 s^−1^. An additional harmonic modulation with an amplitude of 2 nm and a driven frequency of the ac signal of 45 Hz was simultaneously applied on the indenter to perform the continuous stiffness measurement (CSM) for this work [28]. The main purpose of applying the additional harmonic modulation was to avoid the sensitivity to thermal drift. Before each nanoindentation test, it is important to wait until the thermal drift level is below 0.01 nm/s. Furthermore, in order to ensure the statistical significance of the nanoindentation tests, ten indentation locations were performed on each SmBFO thin film, and indentation locations were separated by at least 5 μm.

The *P*-*h* curves were analyzed based on the Oliver–Pharr method [29]. The hardness (*H*) is calculated by dividing the maximum indentation load (*P*_max_) by the projected contact area (*A*_C_) between the indenter and the measured thin film: $H=P_{max}/A_{C}$. Notably, *A*_C_ depends on the shape of the indenter and the contact depth (*h*_C_). For a perfect Berkovich indenter, $A_{C}\approx 24.56h_{C}^{2}$, where $h_{C}=h_{max}-\varepsilon (P_{max}/S)$, *ε* is the geometric constant (*ε* = 0.75 for Berkovich indenter), and *S* is the contact stiffness, which is determined by the initial slope of the unloading curve (*S* = *dP/dh*).

The CSM mode enables the continuous measurement of *S* during loading instead of just at the point of initial unloading. The equation to determine the *S* in a CSM is as follows:(1)$$S={\left[\frac{1}{(P_{max}/h(w))cos\Phi -\left(K_{s}-mw^{2}\right)}-K_{f}^{-1}\right]}^{-1}$$
where *h*(*w*) is the displacement response of the indenter, *Φ* is the phase angle between *P*_max_ and *h*(*w*), *K*_S_ is spring constant at the vertical direction, *m* is the mass of the indenter column, *w* is angular speed equal to 2 π*f* (here, *f* is the driven frequency of the ac signal of 45 Hz), and *K*_f_ is frame stiffness. The *K*_S_, *m*, and *K*_f_ are all constant values for a specific indenter system.

The Young’s modulus is determined by assuming that the contact area remains unchanged during initial unloading. The load–displacement relationship of the initial unloading is related to the stiffness of the measured films and Berkovich indenter and the contact area between the measured films and Berkovich indenter, as described by the following expressions [30]:(2)$$E_{r}=\frac{1}{2\beta h_{c}}\sqrt{\frac{\pi }{24.56}}\left(\frac{dP}{dh}\right)$$
(3)$$\frac{1}{E_{r}}=\frac{1-v^{2}}{E}+\frac{1-v_{i}^{2}}{E_{i}}$$
where *β* is a geometric constant (*β* = 1.00 for Berkovich indenter) and *E*_r_ is the reduced modulus. *E* (*E*_i_) and *v* (*v*_i_) are Young’s modulus and Poisson’s ratio of the measured film (the diamond indenter tip), respectively. Here, *E_i_* = 1141 GPa, *v_i_* = 0.07, and *v_f_* = 0.25 were assumed for all SmBFO thin films.

## 3. Results and Discussion

Figure 1 shows the XRD patterns of SmBFO thin films deposited at various *P*_O2_ of 10 mTorr, 30 mTorr, and 50 mTorr. It is noted that the XRD patterns shown in Figure 1 have been intentionally shifted along the *y*-axis with a constant distance in order to clearly compare the relative intensities and positions of the relevant diffraction peaks. All of the diffraction peaks can be indexed as a single rhombohedral phase of BFO [27], indicating that doped Sm^3+^ ions should have been incorporated well into the BFO crystal structure by replacing the Bi sites. This result is consistent with that reported in Ref. [26], in which it was asserted that the partial substitution of Bi^3+^ ions with slightly smaller Sm^3+^ ions did not show a noticeable effect on the crystal orientation of BFO thin films. Therefore, the SmBFO thin films investigated in the present work can be regarded as homogenous, and the strain effect might be neglected due to the small atomic radius difference between Bi^3+^ (1.15 Å) and Sm^3+^ (1.08 Å).

The average size of crystallites (*D*_s_) can be estimated from the full width at half-maximum (FWHM) of the (110)-peak utilizing the Sherrer’s equation [31]: *D*_s_ =0.9λ/(βcosθ), where *λ* is the wavelength of X-ray radiation (Cu Kα, *λ* = 1.5406 Å), θ is the Bragg angle, and β is the FWHM of the selected diffraction peak. It is noted here that in order to obtain a more precise magnitude of the FWHM, appropriate background subtractions on the diffraction pattern were conducted by running the background traces of the diffractometer used in the present study. As a result, the *D*_s_ values were 20, 22, and 25 nm for the SmBFO thin films deposited at various *P*_O2_ of 10, 30, and 50 mTorr, respectively.

Taking into account the strain effect, the size of crystallites (*D*_WH_) and the local microstrain (ε) of the SmBFO thin films could be calculated by using the Williamson–Hall method [32] as given by the following expression: $\beta \cos\theta =\frac{0.9\lambda }{D_{WH}}+4\varepsilon \sin\theta$. The values of *D*_WH_ and ε were, respectively, obtained from the intercept and the slope of the linear fitting in the plot of βcos⁡θ vs. 4sin⁡θ. Both *D*_s_ and *D*_WH_ values of the present SmBFO thin film and those reported in previous studies are listed in Table 1. The crystallite size calculated using the original Scherrer’s equation systematically gave values smaller than those calculated by using the Williamson–Hall equation, suggesting that the microstrain indeed affected the XRD results significantly. Notice that with increasing *P*_O2_, the intensity of XRD peaks increased and the FWHM decreased, which clearly indicates the enhanced crystallinity of the films (Figure 1). The XRD results evidently show that increasing *P*_O2_ during deposition resulted in increased crystallite size and enhanced crystallinity; thus, it is expected to affect the mechanical properties (e.g., the hardness, Young’s modulus, and mechanical strength) of the SmBFO thin films (see below).

In Figure 2, the SEM images display the surface morphology of the films. All SmBFO thin films exhibit homogeneous and crack-free microstructures consisting of sub-micrometer-dense grains. Moreover, the grain size increases with increasing *P*_O2_, which is consistent with the XRD results (Figure 1) because a higher *P*_O2_ may reduce the deposition rate and thus facilitate grain growth.

Generally, to avoid the substrate effect on the determination of the hardness and Young’s modulus of SmBFO thin films, the indentation depth should not exceed 30% of the film’s thickness [33]. In the present study, the film’s thicknesses were approximately 300 nm, and thus, we applied the maximum penetration depth below 70 nm (corresponding to 23.3% of the film thickness) to ensure that the substrate effect can be neglected. Figure 3 shows the typical CSM *P*-*h* curves for the SmBFO thin films deposited at various *P*_O2_. Akin to the stress–strain curve obtained in the usual mechanical property tests, the *P*-*h* response obtained through nanoindentation contains elastic and plastic deformation due to the compressive indentation. Thus, it can be regarded as indices of the mechanical properties within nanoscale regions. All *P*-*h* curves obtained for the SmBFO thin films appear to exhibit a rather smooth behavior. Indeed, the absence of any discontinuities along either the loading (the so-called “pop-in” event) or unloading (the so-called “pop-out” event) segment indicates that neither the occurrence of fracture nor a pressure-induced phase transition is involved here. That is, this result is in sharp contrast to the observation of a fracture or/and cracking phenomenon in the hexagonal-structured HoMnO_3_ thin films [34] and ZnO thin films [35] and the phase transformation behaviors observed in Si [36,37] and Ge [38] single crystals during nanoindentation.

The hardness and Young’s modulus of all SmBFO thin films are directly determined by the *P*-*h* curves obtained from the CSM results [29], and the results are shown in Figure 4 and listed in Table 1. Briefly, the values of hardness (*H*)/Young’s modulus (E*_f_*) of the SmBFO thin films were 8.2 ± 0.3/114.3 ± 1.2 for the 10 mTorr SmBFO thin film, 7.1 ± 0.1/97.2 ± 2.6 for the 30 mTorr SmBFO thin film, and 6.5 ± 0.2/85.6 ± 4.3 GPa for the 50 mTorr SmBFO thin film. It means that both *H* and *E_f_* values monotonically decreased when *P*_O2_ increased. Figure 5 shows a plot of the *H* versus $D_{WH}^{-1/2}$ data for the SmBFO thin films deposited at various *P*_O2_. Although the crystallite sizes of the SmBFO thin films were relatively small as compared to those of the usual metallic materials, the data closely followed the phenomenological “Hall–Petch” equation [39], as
(4)$$H\left(D_{WH}\right)={H_{0}+k}_{HP}D_{WH}^{-1/2}$$
wherein *H*_0_ and *k*_HP_ are denoted as the lattice friction stress and the Hall–Petch constant, respectively. The dashed line in Figure 5 is the fitting line using the Hall–Petch equation with $H\left(D_{WH}\right)=0.2+44.8D_{WH}^{-1/2}$, which indicates a probable lattice friction stress of 0.2 GPa and where the Hall–Petch constant of 44.8 GPa nm^1/2^ strongly indicates the effectiveness of a grain boundary in hindering of dislocation movements in the SmBFO thin films.

Additionally, the ratio $H^{3}/E_{f}^{2}$ reflects the resistance of the studied material to plastic deformation; the higher $H^{3}/E_{f}^{2}$ value indicates that it is more resistive to plastic deformation under applied stress, and this value also is a reliable indicator of wear resistance. The $H^{3}/E_{f}^{2}$ decreased from 0.0422 to 0.0374 GPa with increasing *P*_O2_ from 10 to 50 mTorr. As is evident from Table 1, the values of *H*, *E_f_*_,_ and $H^{3}/E_{f}^{2}$ of SmBFO thin films decreased in a consistent manner with increasing *P*_O2_ from 10 to 50 mTorr. Notably, the reported *H* and *E_f_* values obtained through nanoindentation for BFO and doped-BFO thin films in this study and in the literature are quite variable (as listed in Table 1), which is attributed to the differences in sample preparation procedures. For example, the 10 at% Sm-doped BFO thin film was deposited under similar conditions (450 °C and *P*_O2_ = 30 mTorr) but obtained a much larger grain size of *D*_S_ = 104 nm [27], and, consequently, the film yielded *H* = 8.8 GPa and *E_f_* = 166.7 GPa. It implies that meaningful comparisons in nanomechanical properties should be always conducted under adequately controllable conditions.

While it was shown by Wang et al. [9] that for a Sm-doped BFO film grown on SrTiO_3_ substrates through PLD, higher substrate temperature can promote the doping of Sm into BFO films and effectively improve the ferroelectric characteristics of the resulting SmBFO films, a substrate temperature of up to 670 °C appears to be not feasible for glass substrates. Alternatively, SmBFO thin films deposited at a higher *P*_O2_ are expected to have a lower concentration of oxygen vacancies, which, in turn, should result in a significant suppression in leakage current and thus substantially enhanced ferroelectric properties [40]. Indeed, by using the Rutherford backscattering analysis, Panchasara et al. had evidently confirmed that the oxygen content in the films was increased substantially when the *P*_O2_ was raised from 30 to 300 mTorr [40], suggesting that, in general, BFO films grown under higher *P*_O2_ environments result in a significant reduction in oxygen vacancies. Moreover, the saturation polarization and coercive field of the BFO thin films were found to increase from 0.12 μC/cm^2^ and 73 kV/cm to 1.5 μC/cm^2^ and 184 kV/cm for BFO thin films grown at *P*_O2_ of 20 and 100 mTorr, respectively. By comparing the results of current–voltage measurements, such enhancements in ferroelectric characteristics were attributed to the lowering of oxygen vacancy concentration and the improved space-charge-limited conduction transport mechanism across the film/substrate interfaces [40].

**Table 1 micromachines-14-01879-t001:** The crystallite size and mechanical properties of some undoped and rare-earth element-doped BFO thin films.

	*D*_s_ (nm)	*D*_WH_ (nm)	*ε* (%)	*H* (GPa)	*E*_f_ (GPa)	$H^{3}/E_{f}^{2}$ (GPa)
BFO thin films [41]	24.5–51.2	―	―	6.8–10.6	131.4–170.8	―
BFO thin films [42]	5.6–10.1	―	―	7.5–7.9	111.1–118.3	―
La-doped BFO ceramics [43]	―	―	―	2.5 ± 0.2	80 ± 10	―
SmBFO_10mTorr ^[#]^	20	33	0.22	8.2 ± 0.3	114.3 ± 1.2	0.0422
(oxygen partial pressure)						
SmBFO_30mTorr ^[#]^	22	41	0.24	7.1 ± 0.1	97.2 ± 2.6	0.0378
(oxygen partial pressure)						
SmBFO_50mTorr ^[#]^	25	52	0.25	6.5 ± 0.2	85.6 ± 4.3	0.0374
(oxygen partial pressure)						

[#]: this study.

## 4. Conclusions

This study reported the thin film growth of 10 at.% Sm-doped BiFeO_3_ (SmBFO) thin films using PLD at various *P*_O2_ of 10, 30, and 50 mTorr and studied their structural, morphological, and nanomechanical properties. All of the films exhibited the typical rhombohedral phase of the BFO structure, in which Sm^3+^ should occupy Bi^3+^ sites, and the crystallite size increased with increasing *P*_O2_ according to the XRD pattern. In addition, the SEM observations showed that all SmBFO thin films presented granular smooth surfaces. From the CSM nanoindentation results, the hardness (Young’s modulus) of SmBFO thin films monotonically decreased from 8.2 ± 0.3 GPa (114.3 ± 1.2 GPa) to 6.5 ± 0.2 GPa (85.6 ± 4.3 GPa) when the *P*_O2_ increased from 10 to 50 mTorr. The hardness of SmBFO thin films appeared to follow the Hall–Petch equation, obtaining a Hall–Petch constant of 44.8 GPa nm^1/2^. This result suggests the effectiveness of a grain boundary in inhibiting the dislocation movement in SmBFO thin films.

## Figures and Tables

**Figure 1 micromachines-14-01879-f001:**
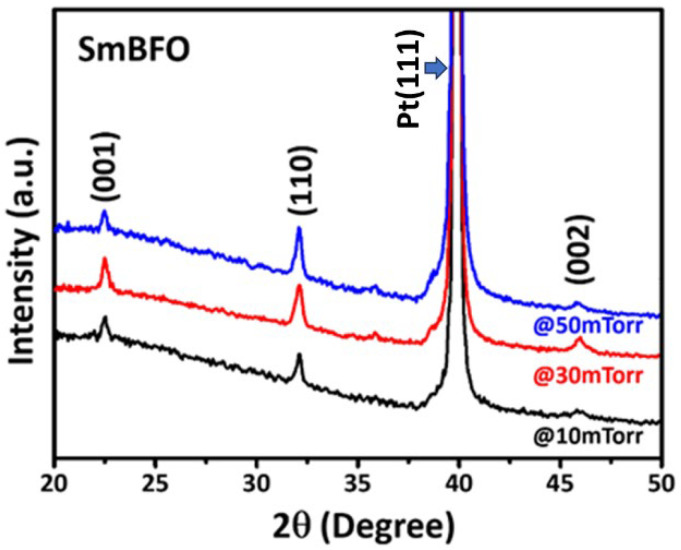
XRD patterns of SmBFO thin films deposited under various oxygen pressures.

**Figure 2 micromachines-14-01879-f002:**
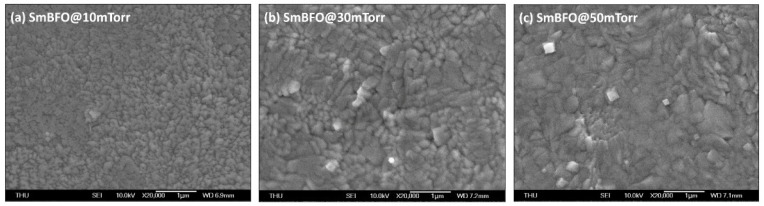
SEM micrographs of SmBFO thin films at various *P*_O2_ of (**a**) 10, (**b**) 30, and (**c**) 50 mTorr.

**Figure 3 micromachines-14-01879-f003:**
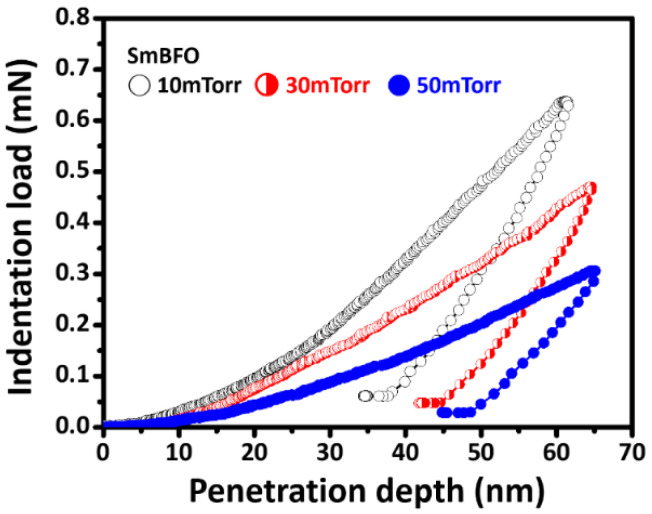
Nanoindentation CSM *P*-*h* curves of SmBFO thin films.

**Figure 4 micromachines-14-01879-f004:**
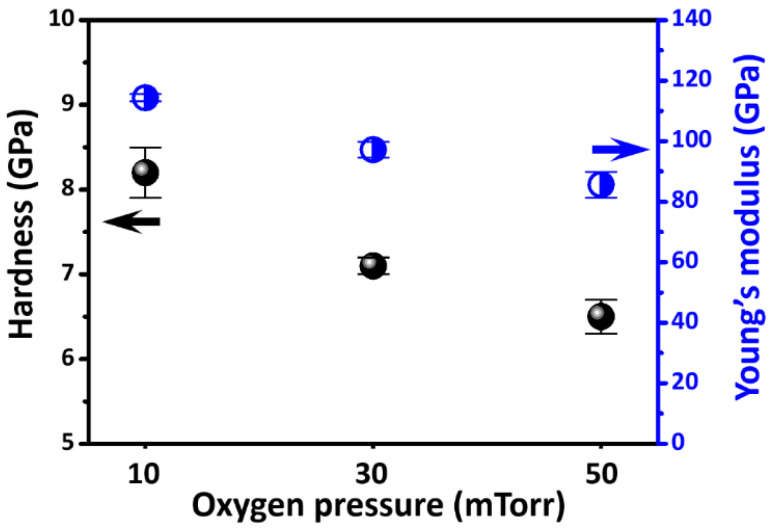
Hardness and Young’s modulus of SmBFO thin films at different *P*_O2_.

**Figure 5 micromachines-14-01879-f005:**
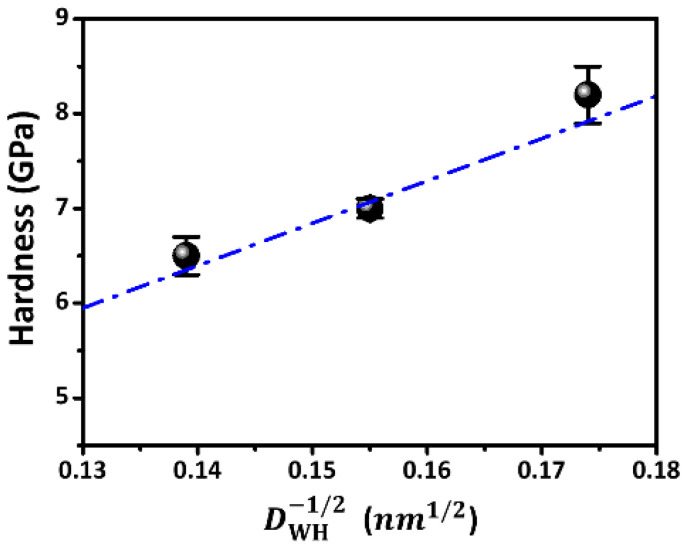
The relation of hardness and crystalline sizes (*D*_WH_) of SmBFO thin films. The dash-dotted line is the fitting line using Hall–Petch equation.

## Data Availability

Not applicable.

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
