# Peer review of "Effects of Oxygen Pressure on the Microstructures and Nanomechanical Properties of Samarium-Doped BiFeO3 Thin Films"

_micromachines, 2023, doi:10.3390/mi14101879_

Round 1

Reviewer 1 Report

The paper is about thin-film samarium (Sm-10%)-doped BiFeO3 (SmBFO) grown on platinum-coated glass. with three different oxygen pressures in the chamber.

Comments and small points as they appear in the text:

  1. 98 units

In Fig. 1, the curves are separated by a constant value, or is this the experimental result? If the shift is intentional, please add an appropriate comment. Why are the curves shown starting at 20 degrees? A background can be observed in all cases. Is this related to some non-periodic products? What is the reflection at 40 degrees? Is it coming from a layer or substrate?

  1. 136: Did you subtract background for the calculations? I can’t find it in the methodology.
  2. 147, you mean that the growth rates slow down with O2 pressure. Did you try to compensate the growth rate with pressure?

In conclusion, it seems that the growth rate (which depends on the O2 pressure) is responsible for the grain size (more time to grow bigger crystallites) on the surface. For that reason, the results are not surprising. Bigger grains have a softer layer and are less elastic. In my opinion, the experiment was well conducted and worth publication.

Author Response

Journal: Micromachines (Manuscript ID: micromachines-2612656)

Title: Effects of oxygen pressure on the microstructures and nanomechanical properties of samarium-doped BiFeO3 thin films

Reply to the Reviewer 1

We would like to thank the reviewers for the critical questions and comments, which indeed help us tremendously in improving our manuscript.

Please kindly find the attached file.

Reviewer 2 Report

The authors investigated the mechanical properties, i.e. hardness, youngs modulus and crystal structure of samarium doped BFOs. The material is interesting due to the co-existence of ferroelectricity and anti-ferromagnetic behavoir. The paper is well-written and well structured. I have two comments on the actual work.

1. The authors only investigated mechnical properties. The author should give an outlook on hwo their finding might affect the film properties in regard of leakage current or coercive field.

2. The authors investigated the crystal structure of the sm-doped BFOs, but an quantitive investigation of the final composition would be desirable and should be prospect of future work. At least in the conclusion the authors should discuss the affect of different partial pressure on film composition and the respective impact on the mechanical properties.

Author Response

Journal: Micromachines (Manuscript ID: micromachines-2612656)

Title: Effects of oxygen pressure on the microstructures and nanomechanical properties of samarium-doped BiFeO3 thin films

Reply to the Reviewer 2

We would like to thank the reviewers for the critical questions and comments, which indeed help us tremendously in improving our manuscript.

Please kindly find the attached file.
